# Reinforcement Learning Finetunes Small Subnetworks in Large Language Models

**Sagnik Mukherjee**    **Lifan Yuan**    **Dilek Hakkani-Tür**    **Hao Peng**
University of Illinois Urbana-Champaign
{sagnikm3,lifan4,dilek,haopeng}@illinois.edu

## Abstract

Reinforcement learning (RL) yields substantial improvements in large language models' (LLMs) downstream task performance and alignment with human values. Surprisingly, such large gains result from updating only a small subnetwork comprising just 5%-30% of the parameters, with the rest effectively unchanged. We refer to this phenomenon as *parameter update sparsity* induced by RL. It is observed across all 7 widely-used RL algorithms (e.g., PPO, GRPO, DPO) and all 10 LLMs from different families in our experiments. This sparsity occurs *without* any explicit sparsity-promoting regularizations or architectural constraints. Finetuning the subnetwork alone recovers the test accuracy, and, remarkably, produces a model nearly identical to the one obtained via full finetuning. The subnetworks from different random seeds, training data, and even RL algorithms show substantially greater overlap than expected by chance. Our analysis suggests that this sparsity is *not* due to updating only a subset of layers; instead, nearly all parameter matrices receive similarly sparse updates. Moreover, the updates to almost all parameter matrices are nearly full-rank, suggesting RL updates a small subset of parameters that nevertheless span almost the full subspaces that the parameter matrices can represent. We conjecture that the this update sparsity can be primarily attributed to training on data that is near the policy distribution; techniques that encourage the policy to remain close to the pretrained model, such as the KL regularization and gradient clipping, have limited impact. Our code is available at https://github.com/SagnikMukherjee/sparsity_in_rl.

## 1 Introduction

Reinforcement learning (RL) (Sutton et al., 1998; Ouyang et al., 2022; Ziegler et al., 2020; Ramamurthy et al., 2023; Sun et al., 2024; Zhou et al., 2025) is an important post-pretraining stage for adapting large language models (LLMs) to solving complex reasoning problems (Lightman et al., 2023; Wang et al., 2025a,c; Cui et al., 2025), alignment with human values (Ouyang et al., 2022; Bai et al., 2022; Dai et al., 2023), and adherence to safety protocols (Mu et al., 2024; Huang et al., 2024; Zhang et al., 2024; Duan et al., 2024). Since these desired behaviors often differ significantly from those of the pretrained model (Ouyang et al., 2022; Bai et al., 2022; OpenAI, 2024), it is often assumed that achieving them requires substantial changes to the model's parameters and therefore full finetuning is widely applied during RL (Cui et al., 2025; HuggingFace, 2025; Liu et al., 2025b; Pan et al., 2025; Zeng et al., 2025).

> **Finding 1**
>
> **RL-induced parameter update sparsity in LLMs:** RL updates only a small subnetwork of a pretrained large language model, leaving the rest of the parameters effectively unchanged.

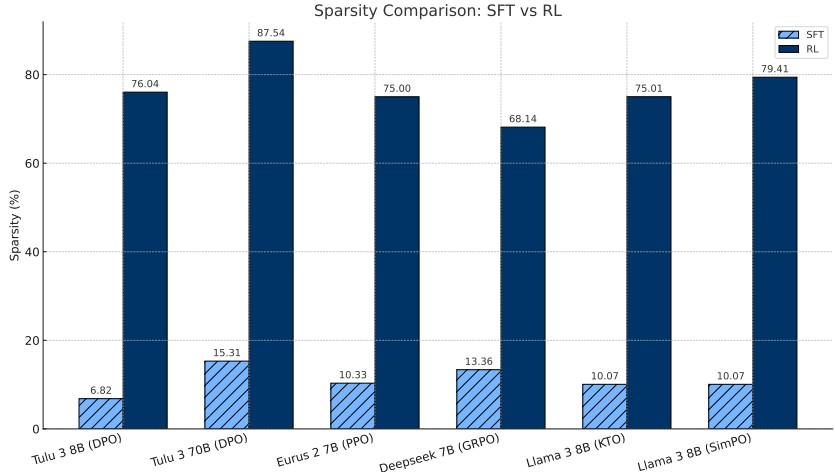

Figure 1: Comparison in accumulated gradients in the SFT stage vs RL stage for popular released checkpoints. SFT stage has accumulated much denser updates, while RL is mostly sparse.

While RL with full finetuning is allowed to update all parameters, does it actually do so? This paper presents surprising findings and answers this question in the negative. Finding 1 is observed in all 7 widely-used RL algorithms studied, namely PPO (Schulman et al., 2017b), GRPO (Shao et al., 2024), ORPO (Hong et al., 2024), KTO (Ethayarajh et al., 2024), DPO (Rafailov et al., 2023), SimPO (Meng et al., 2024) and PRIME (Cui et al., 2025), as well as supervised finetuning with rejection sampling (Xiong et al., 2025), and 10 models in our experiments, with the subnetworks consisting of as little as 5% of the model parameters in some cases (§3). It emerges *without* any explicit sparsity-promoting regularization, architectural constraint, or use of parameter-efficient training or pruning methods. Moreover, we observe a strong consistency among the subnetworks emerged under different random seeds, training data and its order, and even different RL algorithms, suggesting that the pretrained model contains a partially transferable structure that is consistent across varied training conditions (§5).

Interestingly, our experiments with PRIME suggest that approximately 20% of the parameters are consistently updated and make up the subnetwork. An additional 8% receive non-zero gradients during training that cancel out, while the remaining ∼70% parameters remain *untouched* throughout the entire training process. This observation motivates us to articulate the following conjecture:

---

**Conjecture 1**

Fine-tuning only the subnetwork identified at the end of RL training, with all other parameters frozen, produces a model that is nearly identical to the original model that has undergone full finetuning, both in test accuracy and in parameter values.

---

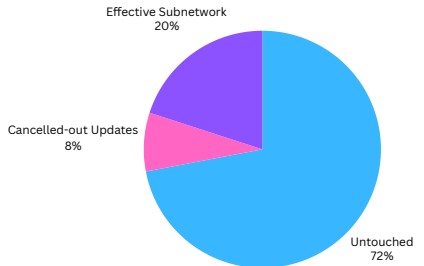

More formally, let $\theta_{\text{full}}$ denote the parameters after full RL finetuning from the initial model $\theta_{\text{init}}$. Define a binary mask $m \in \{0, 1\}^{|\theta_{\text{init}}|}$ where $m_i = 1$ if $(\theta_{\text{init}} - \theta_{\text{full}})_i \neq 0$ and 0 otherwise. We finetune a second model from $\theta_{\text{init}}$ on the same data with the same hyperparameters and number of gradient updates, but, at each step, mask the gradients as $m \odot \nabla_\theta \mathcal{L}(\theta)$ right before the parameter update, so that only the identified subnetwork receives non-zero gradients.

Figure 2: In PRIME, 72% parameters are never updated, 8% have gradients canceling each other out, and 20% constitute the subnetwork that is consistently updated (§6)

Let $\theta_{\text{sub}}$ denote the resulting parameters of this subnetwork finetuning process. Conjecture 1 states that $\theta_{\text{sub}} \approx \theta_{\text{full}}$. We provide supporting evidence for the conjecture on PRIME and DPO in §4.

We find that the updates of RL finetuning do *not* concentrate in specific layers or components of the transformer. Instead, nearly all parameter matrices receive similarly sparse updates (§3). An exception is the layer normalization layers, which receives little to no updates. Moreover, despite the sparsity in the updates, they are almost always full-rank. This suggests that, instead of forcing the

updates to reside in a low-rank subspace as in LoRA (Hu et al., 2022), RL full finetuning updates a small subset of parameters that nevertheless span almost the full subspaces that the parameter matrices can represent.

To better understand the potential reasons for this phenomenon, we conduct a series of experiments in §6. The results indicate that a primary factor is training on data that is near the policy distribution through, e.g., on-policy RL or performing supervised finetuning (SFT) on the same data before RL (Wang et al., 2024a; Cui et al., 2025). Intuitively, it requires less change to the policy distribution when the model learns on a sequence sampled from a distribution close to itself. In contrast, SFT often involves distribution shifts (Zhang et al., 2025) and densely updates the models in our experiments (§3). Other factors like KL-divergence regularization towards the reference model, gradient clipping (as used in PPO, GRPO, and PRIME), online vs. offline RL, all have limited impact on the sparsity of accumulated updates.

Our findings have important implications for the RL fine-tuning stage of LLMs. They suggest that when RL fine-tuning is performed on data closely aligned with the current policy, as is typical in practice, optimization concentrates primarily on a small, consistently active subnetwork, leaving most other parameters effectively inert. Conjecture 1 goes beyond the lottery ticket hypothesis (LTH) (Frankle and Carbin, 2019): not only can the subnetwork, finetuned in isolation, match the performance of the full model in performance as posited by LTH, we show that it also converges to an effectively identical model. These results offer fresh evidence supporting recent findings that RL better preserves the pretrained capabilities compared to SFT (Chu et al., 2025; Setlur et al., 2025), potentially by updating substantially fewer parameters. They also open up new possibilities for more efficient RL training methods that explicitly leverage this update sparsity (Chen et al., 2022).

## 2 Related work and Background

### 2.1 Related Work

The Lottery Ticket Hypothesis (LTH; Frankle and Carbin, 2019) posited that dense neural networks contain sparse subnetworks capable of matching the performance of the full model when trained in isolation. Subsequent extensions to LLMs identified task-specific subnetworks that mitigate catastrophic forgetting without retraining entire models (Panda et al., 2024; Panigrahi et al., 2023; Yadav et al., 2023). Related efforts further discovered sparse subnetworks in pretrained language models crucial for encoding specific knowledge (Marks et al., 2025; Bayazit et al., 2024; Liu et al., 2022). Recent works have also explored exploiting the winning lotteries to improve training efficiency (Chen et al., 2022). While our observation is closely related to LTH, it differs in three core dimensions: (1) LTH identifies winning tickets by pruning, while we study subnetwork that naturally emerge; (2) LTH showed that the final model's performance can be reproduced, we show that, in addition to the performance, the exact same model can almost be recovered; (3) LTH focuses on models trained from scratch, while we focus on finetuning from pretrained LLMs.

Sparse training methods exhibit notable benefits in RL efficiency (Graesser et al., 2022; Sidahmed et al., 2024; Sokar et al., 2022; Tan et al., 2023; Davelouis et al., 2025). Recent studies also employ Low-Rank Adaptation (LoRA) (Hu et al., 2022) in RL and have achieved competitive performance alongside significantly reduced computational overhead (Sidahmed et al., 2024; Wang et al., 2025b). In contrast to approaches like LoRA that explicitly constrain updates to a small number of parameters, we find that fine-tuning the naturally emerging subnetwork can match or even surpass the performance of full-model finetuning. Moreover, despite its sparsity, the updates are nearly full-rank.

### 2.2 Background

We briefly introduce key concepts and notations to be used in onward discussion.

**The sparsity of parameter updates.** Let $\theta^0, \theta^1 \in \mathbb{R}^n$ denote the model parameters before and after finetuning, respectively. We define the **update sparsity** as $\text{sparsity}(\theta^0, \theta^1) := 1 - \left\| \theta^1 - \theta^0 \right\|_0 / n$, where $\left\| \cdot \right\|_0$ counts the number of non-zero elements. It is important to clarify that even the update $\theta_1 - \theta_0$ is sparse, it does not imply that the finetuned model $\theta_1$ is sparse. Since no sparsity is assumed for $\theta_0$, a sparse update can still result in a dense $\theta_1$ if $\theta_0$ is dense.

Table 1: Parameter update sparsity across different RL algorithms. We report sparsity for a suite of open models from Hugging Face. For all models, at least 68.5%—and often much more—of the parameters remain unchanged after RL.

| Algo. | Init Model | RL Model | Update Sparsity | On-Policy | KL | Online |
|---|---|---|---|---|---|---|
| DPO | `Llama-3.1-Tulu-3-8B-SFT` | `Llama-3.1-Tulu-3-8B-DPO` | 81.4 | ✗ | ✓ | ✗ |
| | `Llama-3.1-Tulu-3-70B-SFT` | `Llama-3.1-Tulu-3-70B-DPO` | 95.2 | ✗ | ✓ | ✗ |
| GRPO | `deepseek-math-7b-instruct` | `deepseek-math-7b-rl` | 68.5 | ✓ | ✓ | ✓ |
| | `DeepSeek v3 base` | `DeepSeek-R1-Zero` | 86.0 | ✓ | ✓ | ✓ |
| ORPO | `mistral-7B-v0.1` | `mistral-orpo-beta` | 76.9 | ✗ | ✗ | ✗ |
| KTO | `Eurus-7b-sft` | `Eurus-7b-kto` | 96.0 | ✗ | ✓ | ✗ |
| | `Llama-3-Base-8B-SFT` | `Llama-3-Base-8B-SFT-KTO` | 81.2 | ✗ | ✓ | ✗ |
| PPO | `mistral-7b-sft` | `math-shepherd-mistral-7b-rl` | 80.8 | ✓ | ✓ | ✓ |
| SimPO | `Meta-Llama-3-8B-Instruct` | `Llama-3-Instruct-8B-SimPO` | 86.5 | ✗ | ✗ | ✗ |
| PRIME | `Eurus-2-7b-sft` | `Eurus-2-7B-PRIME` | 77.0 | ✓ | ✗ | ✓ |

Unless otherwise specified, we follow standard practice and consider two `bfloat16` values as equal when their absolute difference does not exceed $10^{-5}$, to account for numerical precision limits.[1] All models in our experiments are in the `bfloat16` data type. Sparsity with different tolerance values can be found in Table 6 in the Appendices.

**Learning from in-distribution data.** We use "in-distribution" to refer to training on data drawn from a distribution that closely matches the current policy. An example is on-policy RL with, e.g., PPO, GRPO, and PRIME, which sample data online from the evolving policy during training. Another way to achieve in-distribution RL is to perform SFT on the same data used for subsequent RL, so that the policy adapts to the data distribution before RL. This is a common practice in off-policy methods like DPO and KTO. On-policy methods inherently train on in-distribution data, and off-policy methods can also do so when the training data closely matches the policy distribution. As we will show later in §6, training on in-distribution data is a primary reason for the update sparsity in RL.

**KL-divergence regularization and gradient clipping in RL.** Two widely adopted techniques to keep the policy from deviating too far from the reference model are KL-divergence regularization and gradient clipping. KL regularization (Schulman et al., 2017a), formally computed as $D_{\mathrm{KL}}(\pi_\theta \| \pi_{\mathrm{ref}}) = \mathbb{E}_{\pi_\theta}\left[\log \frac{\pi_\theta(y|x)}{\pi_{\mathrm{ref}}(y|x)}\right]$, constrains policy shifts. Gradient clipping further stabilizes training by bounding the update norm. Both are widely used in algorithms such as PPO, GRPO, and PRIME. In §6, we show that, counterintuitively, both have limited impact on the update sparsity.

## 3 RL Induces Sparse but Full-rank Updates; SFT Induces Dense Ones

This section aims to answer the following research question

> **RQ1:** *To what extent does RL induce sparse parameter updates and where in the model do these updates occur? How does SFT compare?*

**Setup.** To answer this question, we analyze publicly released model checkpoints on Hugging Face released by the authors. With the exception of models where RL is applied directly to the pretrained base model (e.g., `DeepSeek-R1-Zero`), most models follow a conventional three-stage pipeline: pretraining, supervised fine-tuning (SFT), and RL. We analyze both the RL and SFT stages by measuring the update sparsity between model checkpoints before and after RL or SFT fine-tuning. Our experiments cover `Tulu` 8B/70B (Lambert et al., 2025), `Eurus` 7B (Yuan et al., 2025; Cui et al., 2025), `DeepSeek Math 7B` (Shao et al., 2024), and KTO/SimPO models (Meng et al., 2024).

**Results.** As shown in Table 1, for all RL-finetuned models, 68.5%–96.0% of parameters remain unchanged after RL. This trend holds across different RL algorithms and model families. Particularly, `Deepseek-R1-Zero` presents a update sparsity of 86.0%, regardless of directly training from the pretrained base model, namely RL-Zero (DeepSeek-AI et al., 2025), and large-scale training for over 8K steps. Although exact training configurations are not always available, we observe that within the same model family, larger models tend to show higher sparsity. Importantly, all of these models are

---

[1] E.g., PyTorch uses $10^{-5}$ as the default tolerance for gradient checking.

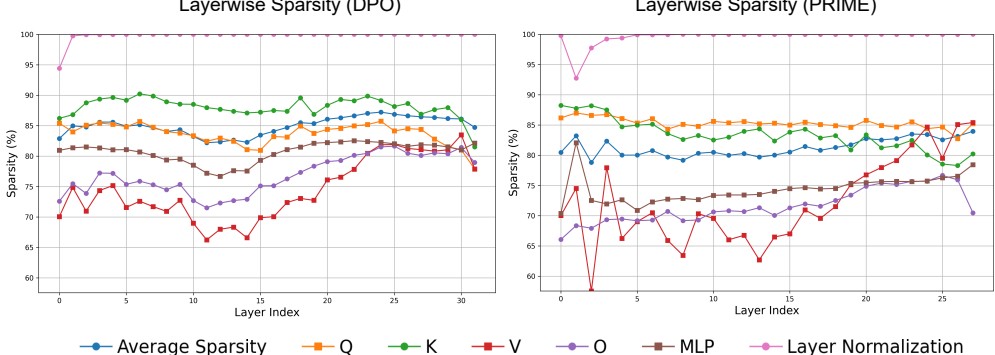

Figure 3: Layerwise and per-parameter-matrix update sparsity for DPO (left) and PRIME (right). All layers are similarly sparsely updated, with the only exception of the layer normalization layers, which receive little to no updates.

trained using full finetuning *without* any sparsity-promoting regularization techniques or constraints. This suggests that the update sparsity emerges naturally.

In contrast, Figure 1 shows that SFT induces dense updates (only 6%-15% sparsity). These results offer fresh evidence supporting recent findings that RL better preserves the pretrained capabilities than SFT (Chu et al., 2025; Setlur et al., 2025), possibly by updating substantially fewer parameters.

> **Takeaway 1**
>
> RL leads to consistently sparse parameter updates (often >70% sparsity) while SFT produces dense updates. This sparsity emerges without regularizations or architectural constraints.

**Almost all transformer layers receive similarly sparse updates.** We next examine how parameter updates in RL are distributed across the model layers and individual parameter matrices (e.g., Q, K, V projections), based on DPO and PRIME models. If updates *were* concentrated in a subset of layers or modules, one could exploit that structure for enhancing the efficiency (Pan et al., 2024). Figure 3 shows layerwise and per-matrix sparsity across the models. The "Average Sparsity" is over each transformer layer, while others correspond to specific parameter

Table 2: Mean ranks of update matrices, as a percentage of maximum possible rank across models after RL finetuning.

| Model and Algo. | Update Rank (%) |
| --- | --- |
| Tulu 8B (DPO) | 99.8 |
| Eurus 7B (PRIME) | 99.5 |
| Llama-3 8B (KTO) | 99.2 |
| DeepSeek Math 7B (GRPO) | 99.4 |

matrices. We observe that parameter updates are distributed across different matrices rather than localized to specific ones. Except for consistently high sparsity in layer normalization layers, most layers exhibit similar sparsity levels. Our results show that sparsity is relatively even across the model. This suggests that recovering the behavior of the fully finetuned model requires updating all layers, albeit with only a subset of parameters in each.

**Updates are sparse but full-rank.** Given the sparsity of RL-induced updates, a natural question is whether these updates are also low-rank. This distinction between low-rank and sparse updates is important: the former would imply that finetuning operates within a subspace, while the latter implies that a small subset of parameters (that can span the full parameter space) are selected to finetune. Notably, while the updates are sparse, a closer inspection reveals that they are nearly full rank (Tab 2). To compute rank, we calculate the average rank of individual update matrices across all layers. We further examine the rank of the update for each layer and parameter matrix, and find that most are full-rank throughout the model. These findings suggest that RL updates are localized to a subset of the parameters that almost span the full subspaces that the parameter matrices can represent, instead of residing in a low-rank subspace.

> **Takeaway 2**
>
> All layers and parameter matrices receive similarly sparse but full-rank updates. While layer normalization parameters are almost never updated.

# 4 Finetuning the Subnetwork Alone Can Reproduce the Full-finetuned Model

Since RL primarily fine-tunes a small subnetwork, we investigate two research questions inspired by but extending beyond the Lottery Ticket Hypothesis (LTH):

**RQ2:** *Can finetuning the subnetwork in isolation recover the performance of the full-finetuned model?*

**RQ3:** *Can subnetwork-only finetuning also recover the exact parameter values produced by full RL finetuning?* This section answers both in the positive.

**Setup.** We follow the procedure described in §1 to obtain two models: one with full finetuning $\theta_{\text{full}}$, and another finetuned on the same data and hyperparameters but updating only the subnetwork $\theta_{\text{sub}}$. We experiment on two very different algorithms DPO, an off-policy algorithm using implicit outcome rewards, and PRIME, an on-policy one with process reward models, to ensure that our conclusion can generalize. We implement DPO with Open-Instruct and PRIME with verl. The exact hyperparameter choices for both can be found in Appendix B. For evaluation, we choose a subset of tasks reported in the original papers for both. For DPO we choose the LSAT (Wang et al., 2022), LogiQA (Liu et al., 2021) splits from AGIEval (Zhong et al., 2024), Math split of MMLU Pro (Wang et al., 2024b). For PRIME, we report results on the MATH500 (Hendrycks et al., 2021) benchmark across difficulty levels. For evaluation in DPO we use olmes.[2]

**Results.** In DPO, 94.0% weights are same between $\theta_{\text{full}}$ and $\theta_{\text{sub}}$; it is 90.5% for PRIME. Notably, for both DPO and PRIME, $\theta_{\text{full}}$ and $\theta_{\text{sub}}$ are 100% identical when using a tolerance of $10^{-4}$ instead of the default $10^{-5}$, indicating that the two models converge to nearly identical parameter values. As shown in Tables 3a and 3b, $\theta_{\text{sub}}$ matches or outperforms $\theta_{\text{full}}$ on all tasks across both algorithms. These results suggest that the parameters outside the subnetwork play little role in the optimization process, and freezing them has a negligi-

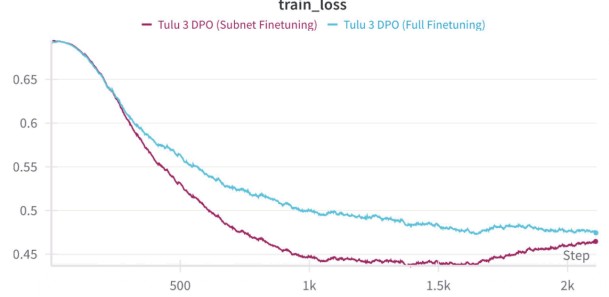

Figure 4: Training loss for training DPO with subnetwork finetuning and full finetuning. Training the subnetwork in isolation consistently causes train loss to be lower.

ble or even beneficial impact on the model's performance. We further observe that the training loss is consistently lower in the subnetwork finetuning setting than full finetuning (Fig. 4). They also provide supporting evidence for our Conjecture 1 in §1. This finding opens up new possibilities for more efficient RL training methods that explicitly leverage this update sparsity (Chen et al., 2022).

Table 3: Test set performance of $\theta_{\text{full}}$ and $\theta_{\text{sub}}$ trained with DPO and PRIME. Training only the subnetwork ($\theta_{\text{sub}}$) can achieve better performance than full finetuning ($\theta_{\text{full}}$). Lvl. indicates the difficulty levels of MATH500.

<table>
<tr><td colspan="4" align="center">(a) DPO</td><td colspan="4" align="center">(b) PRIME</td></tr>
<tr><th>Task</th><th>$\theta_{\text{full}}$</th><th>$\theta_{\text{sub}}$</th><th>$\Delta$</th><th>Lvl.</th><th>$\theta_{\text{full}}$</th><th>$\theta_{\text{sub}}$</th><th>$\Delta$</th></tr>
<tr><td>AGIEval LSAT-AR</td><td>21.3</td><td>24.8</td><td>+3.5</td><td>1</td><td>93.0</td><td>93.0</td><td>+0.0</td></tr>
<tr><td>AGIEval LSAT-LR</td><td>53.1</td><td>54.7</td><td>+1.6</td><td>2</td><td>85.6</td><td>85.6</td><td>+0.0</td></tr>
<tr><td>AGIEval LogiQA-en</td><td>43.5</td><td>45.5</td><td>+2.0</td><td>3</td><td>82.9</td><td>83.8</td><td>+0.9</td></tr>
<tr><td>GPQA</td><td>32.8</td><td>32.8</td><td>+0.0</td><td>4</td><td>71.1</td><td>74.2</td><td>+3.1</td></tr>
<tr><td>MMLU Pro Math</td><td>50.8</td><td>51.6</td><td>+0.8</td><td>5</td><td>40.3</td><td>45.5</td><td>+5.2</td></tr>
<tr><td>**Avg**</td><td>40.3</td><td>41.9</td><td>+1.6</td><td>**Overall**</td><td>69.8</td><td>72.2</td><td>+2.4</td></tr>
</table>

---

[2]Open-Instruct: `https://github.com/allenai/open-instruct`; verl: `https://github.com/volcengine/verl/`; olmes: `https://github.com/allenai/olmes/`

Table 4: Subnetwork overlap varying random seeds, training data, and RL algorithms. Despite these changes, subnetworks show non-trivial overlap compared to random-guessing baselines.

| Variation Axis | Setting | Random | RL Subnetwork | Sparsity |
|---|---|---|---|---|
| Seed | $\mathcal{I}_1$: 42 | $o_1$: 36.7 | $o_1$: 60.5 | 63.3 |
| | $\mathcal{I}_2$: 123 | $o_2$: 36.7 | $o_2$: 60.6 | 63.3 |
| Data | $\mathcal{I}_1$: Tulu Data | $o_1$: 14.6 | $o_1$: 26.7 | 63.3 |
| | $\mathcal{I}_2$: PRIME Data | $o_2$: 36.7 | $o_2$: 67.1 | 85.4 |
| Seed + Data + Algo. | $\mathcal{I}_1$: DPO | $o_1$: 23.0 | $o_1$: 59.1 | 87.1 |
| | $\mathcal{I}_2$: PRIME | $o_2$: 12.9 | $o_2$: 33.2 | 77.0 |

## 5 Consistency of Subnetworks Across Seeds, Data, and Algorithms

This section aims to answer the following research question:

**RQ4:** *How consistent is the RL-updated subnetwork under varying training conditions such as random seed, training data, RL algorithm, and even all of them?*

If the subnetwork remains largely consistent across these variations, it would suggest that the identified subnetwork is not merely an artifact of specific training configuration but a generalizable and transferable structure of the pretrained model.

**Setup.** To quantify the similarity between two subnetworks, we define an overlap metric. Let $s_1$ and $s_2$ denote the sparsity levels of two models, and let $\mathcal{I}_1$ and $\mathcal{I}_2$ be the sets of indices of the updated parameters. The size of the common subnetwork is given by $|\mathcal{I}_1 \cap \mathcal{I}_2|$. One-sided overlap is then $o_1 = |\mathcal{I}_1 \cap \mathcal{I}_2|/|\mathcal{I}_1| = |\mathcal{I}_1 \cap \mathcal{I}_2|/(1 - s_1)$, which quantifies the proportion of $\mathcal{I}_1$ that is covered by the subnetwork $\mathcal{I}_2$, i.e., how well $\mathcal{I}_2$ captures the parameters updated in $\mathcal{I}_1$. Similarly, $o_2 = |\mathcal{I}_1 \cap \mathcal{I}_2|/(1 - s_2)$ quantifies how well $\mathcal{I}_1$ captures the parameters updated in $\mathcal{I}_2$. We compare the observed overlaps $o_1$ and $o_2$ against a random guessing baseline, where a subnetwork is constructed by uniformly selecting the same number of parameters as identified by RL (Appendix E).

We evaluate three settings: (1) varying the random seed alone, (2) varying the training data alone, and (3) changing the seed, data, and the RL algorithm altogether as a stress test. We conduct controlled experiments and all factors not under investigation are the same. Unless otherwise mentioned, all ablations were done with a batch size of 32, trained for one epoch with the base model `Tulu-3-8B-SFT`. When varying the training data, we switch between the Tulu preference dataset[3] and the PRIME rollout dataset[4]. To adapt the rollout dataset to DPO format, we select only positive samples, and pair it with a randomly sampled negative one. When varying the RL algorithm, we train a DPO model initialized from `PRIME-RL/Eurus-2-7B-SFT` and compare it to the `PRIME-RL/Eurus-2-7B-PRIME` model.

**Results.** Table 4 reports our observed overlap. Despite changes in initialization, the resulting subnetworks show substantial overlap—well above the random baseline. For instance, varying the random seed yields overlaps of $o_1 = 60.5\%$ and $o_2 = 60.6\%$. Similar consistency is observed when the training dataset is varied. Even under a stress test, where the data, seed, and RL algorithm are all changed, we still observe notable overlaps of 59.1% and 33.2%. These findings indicate the presence of a subnetwork that is at least partially transferrable to other different settings.

> **Takeaway 3**
>
> For a given base model, we observe substantially higher subnetwork overlap than random guessing across different seeds, training data, and RL algorithms. This suggests the potential of a consistent and at least partially transferrable subnetwork structure across these different training settings.

While the observed subnetwork overlap across seeds, datasets, and training algorithms falls short of 100%, it suggests that partial subnetwork reuse may still offer practical utility. In particular, partial subnetwork reuse could reduce redundant computation across repeated RL runs, such as those in hyperparameter sweeps or ablation studies, by partially reusing the subnetworks. In addition, one

---

[3]`allenai/llama-3.1-tulu-3-8b-preference-mixture`
[4]`PRIME-RL/Eurus-2-Rollout`. It has model generations for math datasets alongside a label for correctness

might be able to reuse part of the subnetwork identified by a cheaper algorithm like DPO and reuse it in more expensive ones like PPO, greatly reducing the training cost without sacrificing performance.

# 6 Why Do the Subnetworks Emerge?

This section answers investigates the following research question:

**RQ5:** *What factors contribute to the update sparsity observed in RL finetuning?*[5]

We investigate the following factors: gradient clipping, KL regularization towards a reference policy, performing SFT prior to RL, and the number of RL update steps. Our investigation suggests that the dominating factor is how close the training data distribution is to the policy's, i.e., whether the training data is in-distribution. Another important factor is the total number of gradient updates.

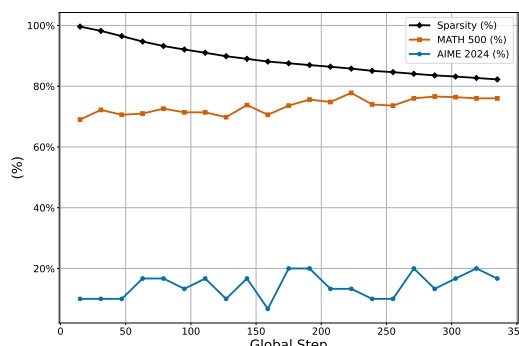

Figure 5: Update sparsity of intermediate checkpoints of a training run of PRIME. We observe that with more training the sparsity slowly decays.

**Gradient clipping and KL regularization.** As discussed in §2.2, gradient clipping and KL regularization are commonly used to keep the policy from deviating too far away from a reference model. Since both mechanisms explicitly suppress large parameter updates, they are natural candidates for contributing factors to the observed update sparsity. To test their impact, we train a GRPO variant using Qwen-2.5-7B-Instruct, comparing models with and without these regularization terms. We find that both configurations exhibit comparable sparsity levels, suggesting that neither gradient clipping nor KL regularization is a primary driver of the update sparsity. In our experiments, the GRPO variant trained with KL regularization achieved a sparsity of 69.8%, while the variant trained without KL regularization reached 68.8%. Further, SimPO removes the KL term by dropping the base policy normalization in DPO, and as reported in 1 SimPO also produces sparse updates, providing further negative evidence for KL.

**Performing SFT before RL.** A common design choice is to perform SFT on the same data as the subsequent RL Ouyang et al. (2022). However, as shown in Table 1, our findings extend to models such as DeepSeek-R1-Zero, which forgoes SFT entirely yet still exhibit high update sparsity. This suggests that SFT is *not* a main contributing factor to the update sparsity in RL finetuning.

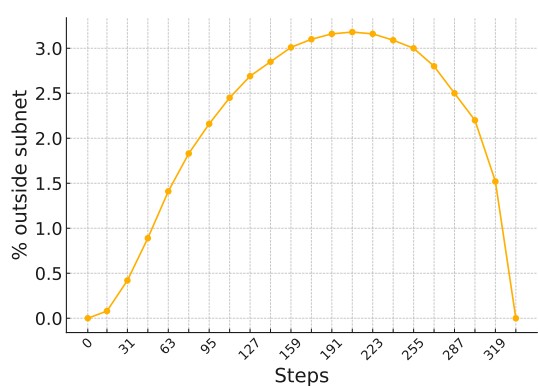

Figure 6: Percentage of updated weights that are outside the final subnetwork across training steps

**Training duration.** It is intuitive that with more gradient steps, a model is expected to drift more from the base model. Figure 5 shows how update sparsity changes during the training of PRIME. As training progresses, update sparsity gradually decreases but eventually shows signs of convergence to around 80%. Notably, `DeepSeek-R1-Zero` under-

---

[5]We conjecture that, for the sake of illustration, if one were to perform backpropagation manually on paper with unlimited numerical precision, the resulting parameter updates would be dense. In practice, however, modern computers rely on floating-point arithmetic with limited precision. As a result, updates with very small magnitudes (e.g., absolute values below $10^{-40}$) cannot be represented and are effectively discarded, hence the empirically observed sparsity of the parameter updates. Importantly, such near-zero updates appear to have negligible impact on model performance, as strong results have been achieved without them. The equivalent question we address in this section is: What factors contribute to these near-zero parameter updates during RL fine-tuning?

goes 8K training steps (numbers from Figure 2 in DeepSeek-AI et al. (2025)) using GRPO, over $20\times$ more than PRIME, but shows a comparable update sparsity (86%). Therefore, we conjecture that training duration's impact on update sparsity is more prominent during early training but gradually decreases as training progresses.

Figure 6 shows the percentage of updated parameters (relative to model size) that lie outside the final subnetwork. This proportion increases during the early stages of training but steadily declines in later stages. This trend suggests that some parameters outside the final subnetwork receive non-zero gradient updates that cancel out. Overall, about 8.5% of parameters that are ever updated during training fall outside the final subnetwork.

While it is possible, though less likely, that all models we study, including `DeepSeek-R1-Zero` (86.0% update sparsity after 8K steps), are severely undertrained, and that the observed update sparsity would diminish with substantially more training. Nonetheless, we question the practicality of this hypothesis since it runs counter to the RL literature arguing against overtraining to prevent overfitting and improve generalization Fu et al. (2019).

**Training on in-distribution data.** Intuitively, when gradients are computed on sequences that the policy already assigns high probabilities to, little update to the parameters would be needed. We evaluated two scenarios: (1) rejection sampling, and (2) DPO on out-of-distribution data by *not* performing SFT prior to RL. Since we've already learned from §3 that DPO with in-distribution data induces sparse updates while SFT (with out-of-distribution data) induces dense ones, this additional experiment can serve as a control group to isolate the factor of training on in- vs. out-of-distribution data.

As shown in Table 5, Our experiments reveal that SFT on in-distribution data produces sparse updates, while DPO with out-of-distribution data produces dense ones. Specifically, performing SFT with `Qwen/Qwen2.5-Math-7B` on rejection sampled in-distribution data yields around 90.0% update sparsity. This is reinforced by the examination of a previous work: RAFT++ (Xiong et al., 2024), which performs supervised finetuning with iterative rejection sampling, yields an update sparsity of 69.4%. In contrast, DPO on out-of-distribution data produces dense updates in `zephyr-7b-beta` models, with a 6.8% update sparsity. These findings suggest that training on in-distribution data could be a major driver of update sparsity in not only RL, but also SFT.

> **Takeaway 4**
>
> We conjecture that training on in-distribution data could be a reason of update sparsity; KL-divergence regularization and gradient clipping have limited impact.

Table 5: RFT indicates rejection-sampling fine-tuning (Touvron et al., 2023; Dong et al., 2023), and RAFT++ is iterative RFT A comparative analysis across SFT and DPO as well as in- vs out-of-distribution training shows that in-distribution consistently produces sparse updates.

| Model | Method | Sparsity (%) | SFT/RL | In-Dist |
|---|---|---|---|---|
| Qwen2.5-Math-7B | RFT | 91.2 | SFT | ✓ |
| Qwen2.5-Math-7B | RAFT++ | 69.4 | SFT | ✓ |
| Llama-3.1-8B-SFT | SFT | 6.8 | SFT | ✗ |
| Llama-3.1-8B-SFT | DPO | 6.8 | RL | ✗ |
| Zephyr-7b-Beta | DPO | 7.7 | RL | ✗ |
| Llama-3.1-8B-DPO | DPO | 81.4 | RL | ✓ |

# 7 Limitations and Future Work

Because RL is computationally demanding, we choose to vary one factor at a time; yet the observed sparsity may actually result from complex interactions among many, an avenue future work should examine. Further, fully controlled experiments are computationally prohibitive, and we thus sometimes resort to resort to public checkpoints. While our experiments focus on language models, it would be interesting to explore the same questions for multimodal and diffusion models. Subsequent research could investigate methods for early identification of the sparse subnetwork and ways to leverage its structure for more efficient learning. Finally, our empirical findings invite a deeper theoretical

analysis, with the goal of uncovering theoretical explanations for the update sparsity in RL. Lastly, while our observations generally hold, there are confounders (for eg. Liu et al. (2025a)). However the confounding reason is not trivial and is worth exploring as part of future work. Further to the best of our observations this phenomenon is largely applicable to all RL settings.

# 8    Conclusion

Our study reveals that RL finetuning in LLMs updates only a sparse subnetwork constituting approximately 5%-30% of total parameters, leaving the rest unchanged. This sparsity emerges without explicit spasity promoting techniques such as regularization or structural constraints. Crucially, finetuning just this subnetwork in isolation reproduces the full model's performance, aligning closely with original parameter values. For a given base model across different seeds, datasets and learning algorithms, a non-trivial portion of the subnetwork remains the same. Our findings highlight that learning from in-distribution samples while training is a key driver of this phenomenon, pointing towards more efficient and effective training strategies in RL-based finetuning of LLMs.

# 9    Acknowledgments

We would like to thank Pavan Jayasinha, Cheng Wang, Abdul Waheed, Shivanshu Shekhar, Alexander Schwing, Gabriel Stanovsky, Roy Schwartz and Gokhan Tur for their valuable feedbacks and suggestions. Further we extend our gratitude to the members of ConvAI and Alta lab groups at UIUC for their thoughtful remarks on our draft. This research used the Delta advanced computing and data resource which is supported by the National Science Foundation (award OAC 2005572) and the State of Illinois. Delta is a joint effort of the University of Illinois Urbana-Champaign and its National Center for Supercomputing Applications.

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

## A  Appendix

| Algorithm | Model (RL Checkpoint) | Tol = 1e-8 | Tol = 1e-7 | Tol = 1e-6 | Tol = 1e-5 |
|---|---|---|---|---|---|
| DPO | allenai/Llama-3.1-Tulu-3-8B-DPO | 76.04 | 76.04 | 76.14 | 81.38 |
| DPO | allenai/Llama-3.1-Tulu-3-70B-DPO | 87.58 | 87.59 | 87.79 | 95.24 |
| GRPO | deepseek-ai/deepseek-math-7b-rl | 68.14 | 68.14 | 68.14 | 68.53 |
| ORPO | kaist-ai/mistral-orpo-beta | 73.16 | 73.18 | 73.23 | 76.94 |
| ORPO | kaist-ai/mistral-orpo-alpha | 50.40 | 50.41 | 50.48 | 53.23 |
| KTO | openbmb/Eurus-7b-kto | 71.78 | 71.79 | 73.14 | 95.98 |
| PPO | peiyi9979/math-shepherd-mistral-7b-rl | 52.45 | 52.47 | 53.21 | 80.77 |
| PPO | PRIME-RL/Eurus-2-7B-PRIME | 75.26 | 75.27 | 75.36 | 77.04 |
| SimPO | Llama-3-Instruct-8B-SimPO | 71.00 | 71.00 | 71.10 | 76.42 |
| SimPO | Llama-3-Base-8B-SFT-SimPO | 79.47 | 79.47 | 79.60 | 86.52 |
| SimPO | Mistral-7B-Instruct-SimPO | 59.37 | 59.40 | 60.31 | 89.07 |
| SimPO | Mistral-7B-Base-SFT-SimPO | 62.58 | 62.60 | 63.56 | 91.44 |

Table 6: Sparsity (%) of parameter updates under different thresholds across RL algorithms and RL checkpoints.

## B  Hyperparameter choices for Gradient Masking experiments

**DPO:** For DPO, we fine-tuned the `LLaMA-3.1-Tulu-3-8B` model using Direct Preference Optimization (DPO) with `bfloat16` mixed-precision and DeepSpeed Stage 3 for memory and compute efficiency across 8 processes. Training uses a sequence length of 2048 tokens with an effective batch size of 128, achieved by setting the per-device batch size to 1 with 16 gradient accumulation steps. A linear learning rate schedule is applied with a peak learning rate of $5 \times 10^{-7}$ and a warmup ratio of 0.1, without weight decay. The model is trained for one epoch on the `allenai/llama-3.1-tulu-3-8b-preference-mixture` dataset.

**PRIME:** For PRIME, We fine-tune `Qwen2.5-Math-7B` using on a mixture of GSM8K and MATH datasets. The training batch size is set to 64. The actor is optimized with a learning rate of $5 \times 10^{-7}$ while the reward model is trained with a learning rate of $1 \times 10^{-6}$. We performed four rollouts are performed per sample. We use gradient clipping of 10.0, and a temperature $\beta$ of 0.05. Training is conducted on for 15 epochs.

## C  Model Checkpoints: SFT vs RL sparsity comparison

**SFT Checkpoints.**  We compare the following base and SFT checkpoints:

- `meta-llama/Llama-3.1-8B` vs. `allenai/Llama-3.1-Tulu-3-8B-SFT`
- `meta-llama/Llama-3.1-70B` vs. `allenai/Llama-3.1-Tulu-3-70B-SFT`
- `Qwen/Qwen2.5-Math-7B` vs. `PRIME-RL/Eurus-2-7B-SFT`

**RL Checkpoints.**  We compare the following SFT and RL-finetuned checkpoints:

- `allenai/Llama-3.1-Tulu-3-8B-SFT` vs. `allenai/Llama-3.1-Tulu-3-8B-DPO`
- `allenai/Llama-3.1-Tulu-3-70B-SFT` vs. `allenai/Llama-3.1-Tulu-3-70B-DPO`
- `PRIME-RL/Eurus-2-7B-SFT` vs. `PRIME-RL/Eurus-2-7B-PRIME`

## D  Training Dynamics

We analyzed intermediate checkpoints of the PRIME model. Notably, our goal is to observe the convergence of the sparsity with training time. Does the sparsity decay with gradient steps ? Does it asymptotically reach a sparsity level of zero or is the convergence to a non-zero point ?

**Experimental setup** We analyze 21 intermediate checkpoints from a training run of the PRIME model. Let the model parameters at these checkpoints be denoted by $\theta_1, \theta_2, \ldots, \theta_{21}$, and let $\theta_{\text{init}}$ denote the parameters of the corresponding base model (i.e., `PRIME-RL/Eurus-7b-sft`). We define the sparsity between the base model to checkpoint $k$ as $sparsity_k = \text{sparsity}(\theta_k, \theta_{init})$, and the sparsity between two checkpoints $i$ and $j$ as $sparsity_{ij} = \text{sparsity}(\theta_j, \theta_i)$.

**Key Findings** Our analysis begins by examining how the $sparsity_k$ evolve with training progress, offering insight into the update patterns.

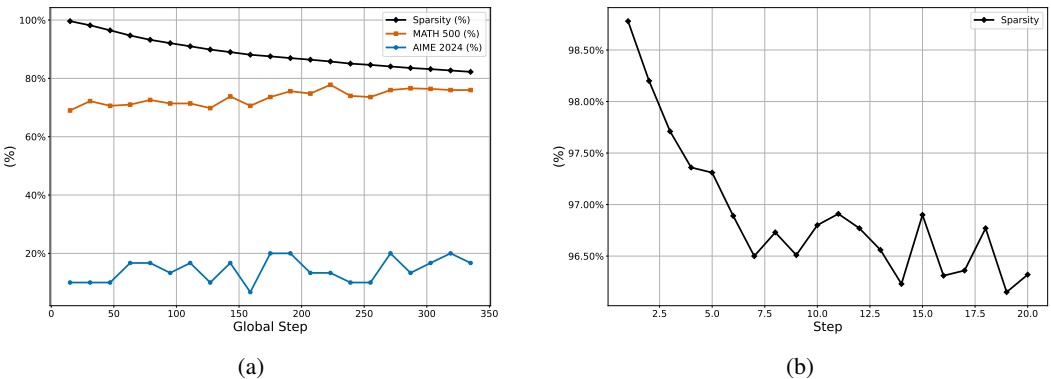

|  (a)  |  (b)  |

Figure 7: Sparsity Analysis of intermediate checkpoints of PRIME (a) shows the sparsity

Figure 7(a) illustrates the sparsity of intermediate checkpoints alongside their accuracy on the MATH500 and AIME2024 tasks. The plot clearly demonstrates that all intermediate checkpoints exhibit non-trivial sparsity. Furthermore, as training progresses, the sparsity converges to a numerically significant asymptote, suggesting that a substantial proportion of weights remain unaffected even after prolonged period of training. In Figure 7(b), we report the sparsity of the $sparsity_{ij}$ for all consecutive checkpoint pairs, i.e., where $j = i + 1$. Notably, in each successive step, on average, only 7% of the weights receive a non-zero gradient update.

# E    Random Guessing baseline

If model 1 has sparsity $s_1$ and model 2 has sparsity $s_2$, the expected overlap is given by: $\frac{(1-s_1)\cdot(1-s_2)}{100}$. Normalizing like earlier, we get $\mathcal{O}_{1,random} = \frac{(1-s_2)}{100}$ and $\mathcal{O}_{2,random} = \frac{(1-s_1)}{100}$. i.e. for random guessing the overlap for model 1 is the density of model 2.

