# OpenReview forum: "Reinforcement Learning Finetunes Small Subnetworks in Large Language Models"
_NeurIPS.cc/2025/Conference — NeurIPS 2025 poster_

### Official Review · Reviewer_jVpM · 2025-06-11

**Clarity:** 3
**Significance:** 3
**Originality:** 3
**Rating:** 4
**Confidence:** 4

**Summary:**

This paper investigates a surprising phenomenon in reinforcement learning (RL) for large language models (LLMs): RL fine-tuning consistently results in sparse parameter updates, affecting only 5–30% of the model’s weights while achieving performance comparable to full-model tuning. This parameter update sparsity emerges naturally—without any explicit regularization—and is observed across a wide range of RL algorithms and LLM architectures.

**Questions:**

1. The authors attribute the observed parameter update sparsity primarily to training on in-distribution data. Could the authors validate this hypothesis by conducting ablation studies using out-of-distribution data within the RL fine-tuning process? Such experiments would help clarify whether the sparsity effect is truly tied to data distribution or influenced by other factors.

2. All experiments in the paper are conducted on transformer-based architectures. Is the observed parameter update sparsity unique to transformers, or does it generalize to other neural network architectures as well? Evaluating this phenomenon beyond transformers would help assess the broader applicability of the findings.

3. The paper suggests that a large portion of parameters receive little to no updates during RL fine-tuning. Could the authors elaborate on why certain gradients consistently remain close to zero across different runs? Is this behavior due to properties of the model, the data distribution, or the RL algorithms themselves? A deeper analysis of this could offer valuable insights into the mechanisms behind update sparsity.

**Ethical Concerns:**

["NO or VERY MINOR ethics concerns only"]

**Final Justification:**

**Review Summary**

This paper investigates a surprising phenomenon in reinforcement learning (RL) fine-tuning for large language models (LLMs): RL fine-tuning consistently produces sparse parameter updates, affecting only 5–30% of the model’s weights, while achieving performance comparable to full-model tuning. This aligns with my own intuition and experience with RL, which tends to be a lighter compared to SFT.

**Concerns and Clarifications**

However, some of the paper’s conclusions could be misleading in their current form. For example, in Line 170, the authors raise the question of whether these updates are also low-rank and then claim they are full rank. Yet, this conclusion is based solely on computing the practical rank, which may lead readers to believe that low-rank updates are impossible in this context. This is not necessarily the case—low-rank approaches could still be viable.

Similarly, while the paper correctly states that SFT updates are dense, prior work [1] has shown that SFT can also yield sparse updates under certain conditions. A more nuanced discussion of this would strengthen the paper and avoid overgeneralization.

**Recommendation**

Despite these issues, I find the core empirical findings compelling and valuable for the community. The contributions outweigh the identified shortcomings. I therefore recommend **Weak Accept**.

[1] Interpreting and Improving Large Language Models in Arithmetic Calculation. ICML 2024.

**Limitations:**

Yes

**Paper Formatting Concerns:**

N.A.

**Quality:**

3

**Strengths And Weaknesses:**

Strengths:

1. Novelty: A key strength of this paper lies in its novel observation of parameter update sparsity during RL fine-tuning of large language models. Unlike prior work that focuses on architectural or algorithmic sparsity, this paper uncovers an intrinsic sparsity pattern that emerges naturally across multiple RL methods and model families—without any explicit regularization.

2. Broad Impact: This insight is both surprising and original, offering a new perspective on how RL updates influence LLMs and opening up potential directions for more efficient fine-tuning strategies. Potential studies on efficient learning or pruning could benefit from this study.

3. Another strength of the paper is its comprehensive experimental validation, covering 7 widely-used RL algorithms (e.g., PPO, GRPO, DPO) and 10 LLMs from different model families. This broad evaluation strongly supports the generality of the observed sparsity phenomenon and adds credibility to the paper’s claims across both methods and architectures.

Weakness:

1. A notable weakness of the paper is that the contributions are largely observational, with limited theoretical or mechanistic analysis. While the discovery of parameter update sparsity is intriguing, the paper does not delve deeply into why this phenomenon occurs or provide formal explanations to support its findings. As a result, the work raises interesting questions but leaves many of them unanswered.

2. Although the updates are sparse, they are also nearly full-rank within each affected matrix, which suggests that their structure is complex and difficult to predict in advance. As a result, it is currently impractical to design algorithms that can exploit this sparsity proactively, limiting the immediate applicability of the findings to algorithmic improvements or efficiency gains.

---

> ### Author Rebuttal · Authors · 2025-07-30
>
> We thank the reviewer sincerely for their time and valuable feedback. We are grateful on their comments on the originality, novelty and potential to influence future work in pruning and efficient learning.  Here we try to address their concerns.
>
> **Weakness 1:**
>
> A notable weakness of the paper is that the contributions are largely observational, with limited theoretical or mechanistic analysis. While the discovery of parameter update sparsity is intriguing, the paper does not delve deeply into why this phenomenon occurs or provide formal explanations to support its findings. As a result, the work raises interesting questions but leaves many of them unanswered.
>
> **Response:**
>
> We thank the reviewer for acknowledging that the observation itself is quite intriguing. The goal of this paper is to highlight a general phenomenon that is observable across RL trained LLM models. While some questions are left unanswered, we believe those are out of the scope, and can be addressed in follow up research.
>
>
> **Weakness 2:**
>
> Although the updates are sparse, they are also nearly full-rank within each affected matrix, which suggests that their structure is complex and difficult to predict in advance. As a result, it is currently impractical to design algorithms that can exploit this sparsity proactively, limiting the immediate applicability of the findings to algorithmic improvements or efficiency gains.
>
> **Response:**
>
> We see immediate efficiency gains in the following possible venues
>
> * One conclusion from our paper is that this subnetwork is consistent (as compared to a random baseline) across variations in seed and dataset. An immediate efficiency gain could be to do a small proxy run to identify the subnetwork. And when searching for better hyperparameters this subnetwork can be reused directly, and this bypasses the need to early detect the subnetworks.
>
> * Further, We conjecture that the curvature of the loss landscape, quantified by, e.g., Fisher information, can potentially reveal interesting insights. A thorough investigation is beyond the scope of this project and is interesting to conduct in future work.
>
>
>
> **Question 1:**
>
> The authors attribute the observed parameter update sparsity primarily to training on in-distribution data. Could the authors validate this hypothesis by conducting ablation studies using out-of-distribution data within the RL fine-tuning process? Such experiments would help clarify whether the sparsity effect is truly tied to data distribution or influenced by other factors.
>
> **Response:**
>
> Thanks for the great question. We aimed to answer this question in Table 5. Row 5 (in table 5) reports update sparsity for an off-policy DPO model, which leads to dense  updates to the base model. This strengthens our conjecture that the nature of in-distribution data causes these sparse updates.
>
> **Question 2:**
>
> All experiments in the paper are conducted on transformer-based architectures. Is the observed parameter update sparsity unique to transformers, or does it generalize to other neural network architectures as well? Evaluating this phenomenon beyond transformers would help assess the broader applicability of the findings.
>
> **Response:**
>
> Thanks for the  interesting question. We focus on transformer-based LLMs because almost all recent LLMs use the transformer architecture and our findings based on it are more practically relevant. As we acknowledged in the “Limitations and Future Work” section and correctly pointed out by the reviewer, it would be interesting to other architectures like diffusion, multimodal LLMs, state-based models; these architectures have fewer readily available open checkpoints that have undergone RL, and a thorough investigation might require more extensive training workload and is therefore left to future work.
>
> **Question 3:**
>
> The paper suggests that a large portion of parameters receive little to no updates during RL fine-tuning. Could the authors elaborate on why certain gradients consistently remain close to zero across different runs? Is this behavior due to properties of the model, the data distribution, or the RL algorithms themselves? A deeper analysis of this could offer valuable insights into the mechanisms behind update sparsity.
>
> **Response:**
>
> Our experiments in section 6 support our conjecture that this is happening due to training data being in-distribution (or on-policy). To validate this, we considered a suite of models that were trained with in-distribution data as well as out-of distribution data. (refer to table 5). Our observations indicated that
> 1.  Rejection sampling (i.e. in distribution SFT) causes sparse updates, while out of distribution DPO causes dense updates.
>
> 2. SFT causes dense updates (which is typically out of distribution) and in-distribution DPO causes sparse updates. Hence we conjecture that the observed sparsity could be driven by the data being in-distribution.

---

> > ### Comment · Reviewer_jVpM · 2025-08-05
> >
> > The reviewer thanks the authors' responses and has carefully reviewed both the replies to my comments and those addressed to other reviewers. I would like the authors to clarify two key questions:
> >
> > 1. Several of my initial concerns require further clarification and let me be more concrete. Regarding parameters that remain unupdated: do you observe that their gradients are consistently zero, or do you employ gradient clipping methods to set them to zero? Additionally, please specify the learning rate used in your experiments and the threshold criteria employed to determine which parameters are considered updated.
> >
> > 2. In your response to Reviewer ks1c, you indicated that SFT also maintains relatively high rank. Could the authors please confirm this observation?

---

> ### Author Response · Authors · 2025-08-05
> **Author Response**
>
> Dear Reviewer,
> Thank you for your time and feedback. Here we try to address the questions that have been raised.
>
> **Question:** do you observe that their gradients are consistently zero, or do you employ gradient clipping methods to set them to zero?
>
> **Response:** In all our experiments, we observed updates to be consistently zero. While PPO uses gradient clipping, our experiments show that it has little impact on the observed sparsity of updates (section 6). Further our observations indicated that the phenomenon of sparse updates hold with DPO and Rejection Sampling, where we do not use gradient clipping.
>
> **Question:** Learning rate for experiments
>
> **Response:** For DPO experiments we used a linear learning rate schedule with a peak learning rate of 5×10−7 and a warm up ratio of 0.1, without weight decay. And in PRIME, the actor is optimized with a learning rate of 5×10−7 while the reward model is trained with a learning rate of 1×10−6. (Appendix B discusses this in more detail alongside other hyperparameters)
>
> **Question:** threshold criteria employed to determine which parameters are considered updated.
>
> **Response:** We consider a parameter as not updated if the magnitude of the update value is less than 1e-5 (Lines 113-116 explains this in more detail). We later ablate this tolerance in table 6 to show that this observation (i.e. the update sparsity) holds across a range of thresholds. (1e-8, 1e-7, 1e-6 and 1e-5).
>
> **Question:** In your response to Reviewer ks1c, you indicated that SFT also maintains relatively high rank. Could the authors please confirm this observation?
>
> **Response:** Yes we observed that the SFT updates are also full rank. However, Despite SFT performing dense updates, the update rank is slightly lower than RL. The fact that RL updates are much sparser makes this observation even more interesting. The detailed response to Reviewer ks1c also contains the corresponding rank of updates in case of SFT.
>
> We hope that this clarifies the reviewer's concerns, and we are happy to provide more details. We will add all these clarifications in our revised version.

---

> > ### Comment · Reviewer_jVpM · 2025-08-05
> >
> > Regarding the final question, the fact that updates in SFT exhibit full-rank characteristics does not preclude the application of low-rank approximation methods. Based on my hands-on experience, SFT models typically results in high-rank parameter updates when conducted without specific regularization constraints. However, parameter-efficient fine-tuning (PEFT) methods such as LoRA, DoRA, and PiSSA can still be effectively employed to achieve low-rank approximations of these updates.
> >
> > This observation raises a critical concern regarding one of the paper's central claims: that updates in reinforcement learning are sparse but high-rank. Given that SFT updates are characteristically full-rank yet can be successfully approximated using low-rank methods, the distinctiveness of the RL update properties requires further justification.

---

> ### Author Response · Authors · 2025-08-05
> **Author Response**
>
> Dear Reviewer,
> We agree with you in that the updates being full-rank does not preclude the application of low-rank approximation methods.
> In fact, our findings **don’t** claim that the full-rank updates in RL imply a low rank approximation can not be done here. We simply raise this as an interesting feature of the sparse subnetwork - that in spite of being quite sparse it is still nearly full rank, i.e. the sparse subnetwork nevertheless span almost the full subspaces. (Line 15, Line 63, Line 177).

---

> > ### Comment · Reviewer_jVpM · 2025-08-05
> >
> > Recent research [1] indicates that SFT can also exhibit sparse and full-rank/low-rank characteristics, which challenges the paper's central premise. The authors in [1] conduct a comprehensive analysis of weight changes during SFT at the attention head level, demonstrating that these updates can be relatively sparse (as illustrated in Figure 2). Their findings reveal that LLMs typically engage only a small fraction (< 5%) of attention heads, which serve a crucial role in directing focus toward operands and operators during computational processes. In this sense, the sparsity in updating is not unique to RL.
> >
> > [1] Interpreting and Improving Large Language Models in Arithmetic Calculation. ICML 2024.

---

> ### Author Response · Authors · 2025-08-07
> **Author Response**
>
> Dear Reviewer,
>
> We thank you for your time and continuous engagement, and this conversation definitely helps us improve the work.
>
> We would like to clarify the following things:
> * The central premise of our work is the observed update sparsity of RL finetuning. The argument on SFT (i.e. SFT updates are dense) is an additional finding and not the central premise of the paper. The main message is that across all major RL algorithms and models a lot of the parameters never underwent any parameter updates at all.
>
> We thank the reviewer for pointing us to this previous paper. We have carefully read it. We believe that their findings, while interesting, are orthogonal to our work and do not affect the value or validity of our contributions.
>
> * [1] studies which attention heads are most active during arithmetic computations. Their Figure 2, plotted changes in activations for attention heads when a data point Xr is replaced with Xc (counterfactual data) for the same model. And this is not directly related to parameter update sparsity.
> * In contrast, our analysis investigates which parameters are actually updated by RL fine-tuning, across a wide array of tasks and models (as compared to only arithmetic computations). Our main result is that RL updates are confined to a small fraction of the total model parameters.
> * Further, While [1] looks at a specific task (arithmetic) while we investigate update patterns induced by RL across general tasks.
>
> Finally, Thank you for your continuous engagement and thoughtful suggestions for improving the paper. We are happy to discuss further in case we missed important context.
>
> [1] Interpreting and Improving Large Language Models in Arithmetic Calculation. ICML 2024.

---

### Official Review · Reviewer_j68b · 2025-06-20

**Clarity:** 2
**Significance:** 3
**Originality:** 3
**Rating:** 5
**Confidence:** 4

**Summary:**

RL has become an important learning algorithm in the post-pretraining stage of LLMs. However, the precise mechanisms through which RL yields its benefits are subject to ongoing investigation. The authors address this question from the lens of parameter update sparsity. They find that in RL finetuning subnetworks emerge that are sparse (less than 30% of the parameters receive non-zero, non-cancelling gradients), but full rank. Using two diverse RL methods - off-policy DPO and on-policy PRIME - they demonstrate that freezing the remaining 70% of the weights when fine-tuning has a negligible or positive effect on benchmark accuracy. According to the authors, this stands in contrast to SFT, which produces dense updates. Emerging subnetworks show considerable consistency across variations in the training pipeline (varying random seeds, preference datasets and RL algorithms), indicating a transferable subnetwork structure. However, when SFT is trained based on rejection sampling (in-distribution data) and DPO on out-of-distribution data, they find that SFT produces sparse updates and DPO dense updates, indicating that a main reason of update sparsity is training on in-distribution data.

**Questions:**

In section 4 the authors answer the two questions - can finetuning the subnetwork in isolation recover the performance of the full-finetuned model? And can subnetwork-only finetuning recover identical parameters as full RL finetuning? - in the affirmative. For this they use DPO and PRIME, covering on and off-policy algorithms. However, the comparison of in vs. out-of-distribution data should be much more at the center of their paper, given the striking difference this makes, and I would like to see this comparison here. Also, given that the paper addresses the difference between SFT and RL, I would be curious to see the same approach applied to SFT.


The paper frames sparsity as a mechanistic hallmark of RL finetuning, but the evidence points instead to the role of distributional alignment. Experiments in §6 show that both RL and SFT can produce sparse updates when the training data is in-distribution. Thus, claims about RL-specific mechanisms should be tempered or reframed accordingly.

If these concerns are addressed, I recommend acceptance.

**Ethical Concerns:**

["NO or VERY MINOR ethics concerns only"]

**Final Justification:**

I appreciated the authors’ thorough rebuttal. I was glad to see that they have taken the concern about overgeneralization seriously and that they planned to revise the framing to clarify that update sparsity is primarily driven by training on in-distribution data, rather than being an intrinsic property of RL algorithms themselves. Their proposed rewordings were appropriate and would improve the scientific accuracy of the claims. I am generally satisfied with the direction of the planned revisions and remain supportive of accepting the paper, maintaining my original score as is.

**Limitations:**

yes

**Quality:**

4

**Strengths And Weaknesses:**

*Strengths*

The paper is timely and investigates the internal mechanisms of RL algorithms and their difference to SFT from an interesting perspective: parameter update sparsity. While this lens has been studied before, the paper focuses on naturally emerging small subnetworks and links these to in- vs. out of distribution training. This, to the best of my knowledge, is a novel insight. The paper furthermore spans a considerable amount of RL algorithms and language models, is methodologically well-conceived and clearly structured.

*Weakness*

The paper’s central claims occasionally overgeneralize what is otherwise a strong empirical result. The evidence suggests that sparsity in parameter updates is not an intrinsic property of RL algorithms themselves, but rather a result of training on data that closely matches the current policy distribution. The authors show that RL with out-of-distribution data does not induce sparsity, and conversely, SFT with in-distribution data does. Yet at several points in the paper, claims are made that attribute sparsity to RL itself.

Below-mentioned sentences overgeneralize and contribute to occasional lack of clarity around what the key driver of sparsity is — algorithmic properties of RL, or data distribution. Such claims should be contextualized more. The paper already provides the corresponding evidence for this contextualization. A more accurate framing would emphasize that update sparsity is primarily driven by distributional alignment, and RL finetuning often happens to operate in this regime. Recasting the finding this way would make the paper’s contribution both more accurate and more broadly applicable.
- “It emerges intrinsically from RL finetuning, without any explicit sparsity-promoting regularization ...”
- “Our findings have profound implications on RL finetuning of LLMs. They suggest that RL concentrates optimization entirely on a small and consistently active subnetwork, while the remaining parameters remain effectively inert.”
- “These results offer fresh evidence supporting recent findings that RL better preserves the pretrained capabilities than SFT.”

---

> ### Author Rebuttal · Authors · 2025-07-30
>
> We sincerely thank the reviewer for their kind comments on the novelty of the finding. We are glad that they found the paper timely as well as methodologically well-conceived and clearly structured. Here we try to address their concerns.
>
> **Weakness 1:**
>
> The paper’s central claims occasionally overgeneralize what is otherwise a strong empirical result....
>
> **Response:**
>
> We thank the reviewer for raising this concern. As the reviewer correctly points out, some statements could unintentionally be interpreted as sparsity being attributed to RL algorithms themselves, rather than RL as a post-training fine-tuning stage for LLMs, which was our intended meaning. We plan to revise relevant statements to address this confusion. Specifically, for those raised by the reviewer, we plan to revise:
>
> * “It emerges intrinsically without any explicit sparsity-promoting regularization, when fine-tuning with on-policy RL algorithms (e.g., PPO, GRPO, REINFORCE) or off-policy algorithms (e.g., DPO) that follow an initial SFT stage on preferred responses, as is common practice.”
>
> * “Our findings have important implications for the RL fine-tuning stage of LLMs. They suggest that when RL fine-tuning is performed on data closely aligned with the current policy, as is typical in practice, optimization concentrates primarily on a small, consistently active subnetwork, leaving most other parameters effectively inert.”
>
> * “These results provide fresh evidence supporting recent findings that RL fine-tuning, usually performed on data close to the current policy, better preserves pretrained capabilities compared to SFT, usually performed on off-policy data.”
>
> **Question 1:**
>
> In section 4 the authors answer the two questions - can finetuning the subnetwork in isolation recover the performance of the full-finetuned model? And can subnetwork-only finetuning recover identical parameters as full RL finetuning? - in the affirmative. For this they use DPO and PRIME, covering on and off-policy algorithms. However, the comparison of in vs. out-of-distribution data should be much more at the center of their paper, given the striking difference this makes, and I would like to see this comparison here. Also, given that the paper addresses the difference between SFT and RL, I would be curious to see the same approach applied to SFT.
>
> **Response:**
>
> This is a great question - and actually the distinction between the on-distribution vs out-of-distribution nature of the data is quite important. However, we would like to highlight that in our experience, an out-of-distribution training does not give rise to sparse updates. And hence finetuning the subnetwork alone induced by RL on out-of-distribution data would require finetuning almost the full model, which we believe we recover full finetuning’s performance and parameter values in a trivial way. Similarly, SFT also causes dense updates to the base model, and hence the subnetwork finetuning experiment will require us to finetune almost the entire model.
>
> **Question 2:**
>
> The paper frames sparsity as a mechanistic hallmark of RL finetuning, but the evidence points instead to the role of distributional alignment. Experiments in §6 show that both RL and SFT can produce sparse updates when the training data is in-distribution. Thus, claims about RL-specific mechanisms should be tempered or reframed accordingly.
>
> **Response:**
>
>
> We appreciate this thoughtful comment. We will definitely make the necessary changes (as already highlighted in our response to weakness 1) such that these concerns are appropriately addressed.

---

> > ### Comment · Reviewer_j68b · 2025-08-04
> > **Thanks for the thoughtful rebutall!**
> >
> > I appreciate the authors’ thoughtful rebuttal. I am glad to see that they have taken the concern about overgeneralization seriously and plan to revise the framing to clarify that update sparsity is primarily driven by training on in-distribution data, rather than being an intrinsic property of RL algorithms themselves.
> > The proposed rewordings are appropriate and improve the scientific accuracy of the claims. I also appreciate the clarification on why subnetwork-only finetuning was not applied to out-of-distribution data or to SFT — the explanation is reasonable and supports the current experimental design.
> > While I would still encourage the authors to ensure that the distributional alignment insight is reflected not only in isolated clarifications but also in the overall framing of the paper, I am satisfied with the direction of the planned revisions.
> > Assuming these changes are implemented as outlined, I remain supportive of accepting the paper and maintain my original score as is.

---

> > > ### Author Response · Authors · 2025-08-05
> > > **Author Response**
> > >
> > > We are glad the reviewer found the responses thoughtful and helpful. We sincerely thank them for their time and valuable feedback.

---

### Official Review · Reviewer_ks1c · 2025-06-23

**Clarity:** 1
**Significance:** 3
**Originality:** 3
**Rating:** 4
**Confidence:** 4

**Summary:**

This work provides an empirical study revealing that RL training for LLMs updates only a sparse subset of the full model, leaving the rest unchanged. They conducted numerical verification to compare the difference between open-source models w/ and w/o RL training. Besides, they conducted experiments to compare full-parameter RL training and subnet RL training; their results show subnet training can achieve similar results on many benchmarks and even better performance than the full-parameter training. They conduct ablation studies to investigate the cause for this phenomenon and attribute that training on in-distribution data is a major reason of sparsity.

**Questions:**

Please see the above section.

**Ethical Concerns:**

["NO or VERY MINOR ethics concerns only"]

**Final Justification:**

I adjust my score to positive as the authors provide missed experimental details and explain the effect of weight decay in their response.

**Paper Formatting Concerns:**

Some important experimental details required by the checklist are missed.

**Quality:**

2

**Strengths And Weaknesses:**

Pros.
1. The analysis experiments comparing the difference between LLMs w/ and w/o RL training are reasonable, e.g., comparing 'Llama-3.1-Tulu-3-8B-SFT' and 'Llama-3.1-Tulu-3-8B-DPO'. They also considered different numerical tolerances from 1e-5, 1e-6, 1e-7, 1e-8, which also makes sense.

2. Then they conducted experiments with corresponding subnet of full parameters for RL training and achieved similar or even better performance.

3. They conducted analysis experiments to investigate the reason behind the phenomenon. They analyze different factors: performing SFT before RL, training duration, in-distribution data, KL-divergence regularization, and gradient clipping. The ablation study covers extensive factors; the comparison results between in-distribution and out-of-distribution are significant.

Cons:

1. As this is mainly an empirical study, there are many experimental setups that should be further clarified. e.g., What is the optimizer used in each experiment? The checklist section: Experimental setting/detail is marked 'yes', but clearly, I didn't find it in the full content or appendix.

2. If the authors use adaptive optimizers, then the updates are different from gradients. For example, in line 56, the author mentioned they masked the gradient. But it is not clear what is done.
Do authors mask it before computing the slot variables like momentum or after? Could authors specify how to mask the update for a subnetwork with an optimizer?

3. Some claims lack enough support, like the full rank of updates.
Again, for table 2 and lines 170-179, the authors use updates.
  -  Could you provide more details about the update rank?
  -  How do you compute the rank, what model parameters did you consider? Which tools or methods did you use to calculate the rank? How many layers or which layers did you consider?
  -  Did you consider the embedding layer or the final language modeling head layer (which is large in size).
  -  Did you compare the corresponding rank of updates for SFT?

4. Why don't you consider weight decay as an ablation factor? Is SFT training with weight decay like using AdamW, and your experiments don't?

---

> ### Author Rebuttal · Authors · 2025-07-30
>
> We appreciate the reviewer’s insightful comments on the strengths and weaknesses of the paper. Here we try to address their concerns.
>
> **Weakness 1:** As this is mainly an empirical study, there are many experimental setups that should be further clarified. e.g., What is the optimizer used in each experiment? The checklist section: Experimental setting/detail is marked 'yes', but clearly, I didn't find it in the full content or appendix.
>
> **Response:**
>
> * Thanks for highlighting this. Since there are ablation experiments throughout the paper, it’s hard to have one consolidated Experimental Setting. We provided detailed descriptions for hyperparameters in **Appendix B**, which is followed throughout the paper, unless otherwise mentioned. Here is a more thorough description of the hyperparameters used, which we will provide in our revised version.
> * For all the experiments in the paper, the optimizer is the **AdamW optimizer** with consistent hyperparameter setting with the original Tulu models as well as the PRIME models. (details in [1] and [2]). Please note that this is the standard setting in most open models and frameworks. (Open-instruct [3], verl[4], GRPO [5] etc. )
>   * Specifically for all DPO experiments we use AdamW with weight_decay=0, learning rate = 5e-7. And for PRIME, we use AdamW with a constant learning rate of 5e-7 for the policy model and 1e-6 for the PRMs (weight decay set to 0).
> * Summarising, for section 4 the experimental details are in appendix B, where we provided details for the gradient masking experiment. For experiments in section 5 all hyperparameters are kept the same as in section 4. And other changes specific to the experiment are discussed in place. The seed and dataset experiments are done with DPO, and the settings are the same as in appendix B. The Seed + Data + Algo experiments are also done with default configurations of DPO and PRIME as discussed in appendix B. We will add a detailed section in the revision. Similarly for section 6, all ablations were done with keeping the same hyperparam settings as in 6, but only ablating the factor mentioned.
> * We will make a detailed subsection in Appendix to address each section’s hyperparameter choices separately.
>
>
> **Weakness 2:** If the authors use adaptive optimizers, then the updates are different from gradients. For example, in line 56, the author mentioned they masked the gradient. But it is not clear what is done. Do authors mask it before computing the slot variables like momentum or after? Could authors specify how to mask the update for a subnetwork with an optimizer?
>
> **Response:** Concretely, on every backward call we register a tensor‑level hook (param.register_hook) that zeroes the entries outside the sub‑network. PyTorch fires this hook immediately after autograd has produced the raw gradient and before the tensor is stored in param.grad; the masked value is therefore what AdamW uses to compute its first‑ and second‑moment estimates. Effectively, this forces that all parameters outside the subnetwork to always receive zero updates
>
>
> **Weakness 3:** Some claims lack enough support, like the full rank of updates. Again, for table 2 and lines 170-179, the authors use updates. Could you provide more details about the update rank? How do you compute the rank, what model parameters did you consider? Which tools or methods did you use to calculate the rank? How many layers or which layers did you consider? Did you consider the embedding layer or the final language modeling head layer (which is large in size).
>
> **Response:**
>
> Thank you for your detailed question. Here we have detailed our approach in computing the rank as reported in the paper.
>
> *Short summary response*
>  * tool used for computing rank is torch.linalg.matrix_rank
>  * Layers considered - all layers except layer norm, including embedding and language modeling head
>  * Method to compute rank - detailed below
>
> Here is the detailed approach to how we computed the reported ranks: In order to compute rank,
> * We first compute the element wise delta matrices for all sublayers (please note that we considered all layers and sublayers) - This includes embedding, q_proj, k_proj, v_proj, FFN, including the lm_head. Which results in a matrix containing the element wise update values for each sublayer. We don't compute ranks for the layernorm weights, since the weight vectors are 1 dimensional ( These amount to only 0.04% of all model weights ).
> * Next compute rank for all of these matrices. For each matrix we compute their matrix ranks separately with built-in function  torch.linalg.matrix_rank (this rank is computed using Singular Value Decomposition).
> * Next we compute the rank as a fraction of the maximum possible rank for the respective matrices
> * We report the mean of this fraction (across all sublayers) in the paper.
>
>
> **Weakness 4:** Did you compare the corresponding rank of updates for SFT?
>
> **Response:**
>
> The corresponding rank of updates in SFT is captured in the table below. Despite SFT performing dense updates, the update rank is slightly lower than RL.  The fact that RL updates are much sparser makes this observation even more interesting. (Note: The following ranks for the SFT stage is computed in the same approach detailed above)
>
>
> | Model  family        | SFT rank | RL rank |
> |----------------|----------|---------|
> | tulu 8b        | 99.75    | 99.8    |
> | Eurus 7b       | 97       | 99.5    |
> | Llama 3 8b KTO | 98       | 99.2    |
>
>
> **Weakness 5**: Why don't you consider weight decay as an ablation factor? Is SFT training with weight decay like using AdamW, and your experiments don't?
>
>
> **Response:** Thank you for your thoughtful question. For the tulu series of models, both the SFT stage [6] as well as the DPO stage [7] use a  weight decay of 0; while the SFT yields dense updates and the DPO stage yields sparse ones. This provides evidence that weight decay is not a major factor in terms of the sparsity of the updates and therefore we do not think it is a  contrasting factor to ablate. Further to validate our conjecture we set weight decay to 0 for the policy model as well as the reward model in PRIME and trained for 150 steps on GSM8k and MATH (please note due to the time constraint of the review period, its infeasible to train for more steps). And the resulting model had 81.6% sparse updates. This further strengthens our conjecture that weight decay is not the root cause for sparse updates.
>
>
>
> [1] Tulu 3: Pushing Frontiers in Open Language Model Post-Training
>
> [2] Process Reinforcement through Implicit Rewards
>
> [3] open-instruct GitHub repo - open-instruct/open_instruct/dpo_tune_cache.py
>
> [4] Verl GitHub repo - verl/workers/fsdp_workers.py
>
> [5]deepseekmath: Pushing  the Limits of Mathematical Reasoning in Open Language Models
>
> [6] open-instruct GitHub repo - scripts/train/tulu3/finetune_8b.sh
>
> [7] open-instruct GitHub repo - scripts/train/tulu3/dpo_8b.sh

---

> > ### Comment · Reviewer_ks1c · 2025-08-01
> >
> > Thanks for the response. The missed optimizer detail is supplied, and the explanation for weight decay looks reasonable to me. I will adjust my rating to positive accordingly. The findings in the paper are interesting and somehow surprising to me. While the weight decay is not used for your RL training, I am still curious about what the impact of applying weight decay is to the full model. Will it help the performance or not, or is your current work enough to deliver a takeaway: Don't use weight decay in the RL training? The answers to these questions are beneficial to the foundation model community.

---

> ### Author Response · Authors · 2025-08-01
> **Author Response**
>
> Dear Reviewer, we are glad that you found the work interesting and thank you for raising the scores.
>
> Regarding the clarification you asked for:
>
> We would like to clarify that, our ablations on factors such as KL divergence, gradient clipping or weight decay are with a sole purpose to check if they affect the phenomenon of sparse updates. The current work is not a recommendation of the form "Don't use weight decay in the RL training", but rather a claim "weight decay is not the root cause of sparse updates".
>
> A detailed analysis including final performance would be quite interesting to see, but that seems out of scope for the purpose of this paper.
>
> Hope this clarifies your question.

---

> > ### Comment · Reviewer_ks1c · 2025-08-05
> >
> > Regarding the experiment you mentioned involving Tulu 8B, I’d like some clarification on the SFT dataset and the learning rate used. Open-Instruct’s default learning rate for SFT training is 2e-5, which is substantially higher than the rate used in your RL training (`2e-5` vs `5e-7`). Additionally, the SFT mixture (allenai/tulu-3-sft-mixture) contains nearly three times the amount of data compared to the DPO mixture (allenai/llama-3.1-tulu-3-8b-preference-mixture). These discrepancies raise some concerns regarding the training setup and its implications. Could you elaborate?

---

> > > ### Comment · Reviewer_jVpM · 2025-08-06
> > >
> > > The reviewer will second this. Considering the huge difference between SFT (lr: 2e-5) and RL (lr: 1e-7), it is possible the sparsity is caused by the learning rate.  It is necessary to exclude the influence the learning rate, especially when you set a threshold to determine which parameters are considered to be fixed.

---

> ### Author Response · Authors · 2025-08-08
> **Author Response**
>
> Dear reviewers,
> Thanks for your continuous engagement. We would like to add our clarification regarding the recent points raised here.
>
> **Concern:** SFT mixture contains 3x more data.
>
> **Response:** Thanks for the great point. The experiments in our paper closely followed the setting by the Tulu 3 paper [3]. Following your suggestion, we tried a controlled experiment performing both SFT and DPO for the same 2000 steps on the same amount of data (since the Tulu 8b DPO model trains for 2000 steps). Comparing this SFT model with the DPO model controls for the number of gradient steps.
> Our observations are as follows:
>
> * SFT for 2000 steps - sparsity = 17%
> * DPO for 2000 steps - sparsity = 81.55%
>
> We observe that even when SFT is done for the same number of gradient steps and data points, SFT updates are substantially denser than DPO, aligning with the observations in the paper. These details will be added to the revised version.
>
> **Concern:** Open-Instruct’s default learning rate for SFT training is 2e-5
>
> **Response**
> With all due respect, we beg to differ. The learning rate that open-instruct uses for training the SFT model is 5e-6, and not 2e-5. This can be verified with the Tulu 3 paper (section 4.3 "SFT Recipe and Analyses" as well as table 11 in [3]) as well as the scripts/train/tulu3/finetune_8b.sh (in [4]).
> Further our experiments follow the standard practice, and in DPO/RLVR it is recommended to keep the learning rate low since otherwise the KL loss explodes and harms learning (refer to footnote in the tulu 3 paper in page 36)
>
> We would also like to add the following clarifications:
> * for DPO it is well established that a high learning rate causes severe degradation to the base model
>     * [1] - page 5 - "with a slightly larger learning rate like 5e − 6 to 1e − 5, the DPO optimized model starts to generate repeated token"
>     * [2] - “A large learning rate (e.g., 1e-5) can significantly degrade performance, causing the model to produce incoherent sentences”
>     * The same holds for RLVR (page 36 footnote 18 in tulu paper, [3])
>
> These two observations are further confirmed by our own experiments, where we are not able to successfully train a DPO model with high learning rate without severe degradation in the model.
>
> We will definitely add these to the paper since these are important clarifications, and are happy to engage in further conversation and clarify any concerns further.
>
> [1] Minor DPO reject penalty to increase training robustness
>
> [2] princeton-nlp /SimPO - github repository
>
> [3] Tülu 3: Pushing Frontiers in Open Language Model Post-Training
>
> [4] allenai/open-instruct - github repository

---

> > ### Comment · Reviewer_ks1c · 2025-08-08
> >
> > Thanks for the further clarification; I strongly recommend the authors add these new experiments and results in the final version of the paper.

---

### Official Review · Reviewer_2DPM · 2025-06-27

**Clarity:** 3
**Significance:** 3
**Originality:** 3
**Rating:** 5
**Confidence:** 4

**Summary:**

This analysis paper studies how fine-tuning a pre-trained LLM via RL changes the weights of the model. Their analysis focuses particularly on parameter sparsity across 7 RL algorithms and 10 LLMs. The authors find that RL only changes a small subset of all parameters (5-30%). In contrast, SFT results in dense updates. They present a number of analyses on what parameters exhibit high sparsity, the rank of the updates, subnetwork overlaps between individual runs, the effect of update constraints (grad clipping, KL), and the effect of in-distribution data on sparsity.

**Questions:**

1. What are your thoughts on how the subnetworks could be identified prior to or at the beginning of the fine-tuning process?
2. How does the optimizer choice affect the resulting sparsity?
3. How does the downstream performance change when you initialize the RL fine-tuned checkpoint with the base model weights for the specific weights that you identified as unchanged (i.e., &lt; 1e-5)? It is possible that the small differences nevertheless impact generation.

**Ethical Concerns:**

["NO or VERY MINOR ethics concerns only"]

**Final Justification:**

The additional analyses on the choice of optimizer and weight resetting are useful additions to the main message of the paper. I am happy to maintain my original score.

**Limitations:**

yes

**Quality:**

3

**Strengths And Weaknesses:**

Strengths:



1. The findings in this paper are well-presented, insightful, and can be interesting for future works on RL fine-tuning.
2. They rely on the concept of update sparsity to illustrate how much/little the parameters change, which is a simple concept but demonstrates their findings well.
3. They evaluate several RL algorithms and models and find that the phenomenon they study persists throughout. This suggests that their findings are generally applicable, rather than an artifact.

Weaknesses:



1. While the paper provides extensive empirical evidence, there is no theoretical analysis of why the subnetworks arise. However, I understand the difficulty of this.
2. It is not investigated and therefore unclear how the choice of the optimizer affects the sparsity ratios.

---

> ### Author Rebuttal · Authors · 2025-07-30
>
> We thank the reviewer for their time and valuable comments. It is great to read that they found the paper well-written and the analysis insightful. Here we try to address their questions
>
> **Weakness 1:**
>
> "While the paper provides extensive empirical evidence, there is no theoretical analysis of why the subnetworks arise. However, I understand the difficulty of this."
>
> **Response:**
>
> We appreciate the reviewer acknowledging that a detailed theoretical analysis is beyond the scope here. The goal of this paper is to highlight a general phenomenon that is observable across RL trained LLMs. We do think the observation itself is  intriguing and important, and makes a valuable contribution alongside the attributes of it that we study in the form of RQs in our paper.
>
> We conjecture that the curvature of the loss landscape, quantified by, e.g., Fisher information, can potentially reveal interesting insights. A thorough investigation is beyond the scope of this project and is interesting to conduct in future work.
>
> **Weakness 2:**
>
> "It is not investigated and therefore unclear how the choice of the optimizer affects the sparsity ratios."
>
> **Response:**
>
> Our observations from the standard training recipes of popular open sourced models are that they often use the same AdamW optimizer for the SFT stage [3] as well as the alignment stage [4] (causing sparse updates in RL and dense updates in SFT). And hence the optimizer did not stand out as a candidate causing sparse updates in RL.  To validate this conjecture, we experimented with the RMSProp optimizer for DPO (keeping other training configurations the same), Our observations show that it yields 81.85% sparse updates, aligning with our hypothesis.  We will include this in the appendix section of our revision.
>
>
> **Question 1:**
>
> What are your thoughts on how the subnetworks could be identified prior to or at the beginning of the fine-tuning process?
>
> **Response:**
>
> This is a great question, and we did some initial explorations here. One approach here could be to do a small proxy run to identify the subnetwork. Our initial experiments there revealed that within the first 40% of steps, 58% of the subnetwork can be recovered. This happens because the subnetwork evolves over training. Further, we conjecture that Fisher information could potentially be an effective way to identify the subnetwork prior to training [1]; techniques from the lottery ticket hypothesis [2] literature can also be promising. A thorough investigation is definitely interesting and of practical benefits, which we plan to explore in future work.
>
> **Question 2:**
>
> How does the optimizer choice affect the resulting sparsity?
>
> **Response:**
>
> The response to the weakness 2 can be referred to here.
>
> **Question 3:**
>
> How does the downstream performance change when you initialize the RL fine-tuned checkpoint with the base model weights for the specific weights that you identified as unchanged (i.e., < 1e-5)? It is possible that the small differences nevertheless impact generation.
>
> **Response:**
>
> Thanks for the interesting question, and in order to do a sanity check we set the DPO model’s updates <1e-5 to 0. And observe no change in downstream performance. Indicating these are indeed noisy updates and have little impact on the performance. The downstream task performance after resetting the weights are shared below.
>
> | Task                    | Tulu 8b | Reset weights |
> |-------------------------|---------|--------------|
> | agi eval:: lsat ar      | 21.3    | 21.3         |
> | agi eval:: lsat lr      | 53.13   | 53.13        |
> | agi eval logiqa_en      | 43.47   | 43.62        |
> | GPQA                    | 32.81   | 32.14        |
>
>
>
> [1] Training Neural Networks with Fixed Sparse Masks
>
> [2] The Lottery Ticket Hypothesis: Finding Sparse, Trainable Neural Networks
>
> [3] open-instruct GitHub repo - open-instruct/open_instruct/finetune.py
>
> [4] open-instruct Github repo - open-instruct/open_instruct/dpo_tune_cache.py
>
> (We can not link the last two references due to this year's rebuttal policy on not using links)

---

> > ### Comment · Reviewer_2DPM · 2025-08-04
> >
> > Thank you for the clarifications and providing the additional analyses on the choice of optimizer and weight resetting. I think they are useful additions to support the main message of your paper. I am happy to maintain my original score.

---

> > > ### Author Response · Authors · 2025-08-05
> > > **Author Response**
> > >
> > > We sincerely thank the reviewer again for their time and valuable feedback.

---

### Note · Authors · 2025-08-13

Dear Reviewers, thanks for your valuable feedback and clarifying questions. We sincerely thank the ACs, SACs, PCs for your service and your help in improving our paper through discussion with the reviewers.

To summarize, three reviewers had scored the paper 5 out of 6. We were encouraged to see reviewer Ks1c, who had scored it 2, saying that they will increase the score to positive in light of our responses.
To summarize individual reviews,

* Reviewer 2dpm found the work insightful, with findings generally applicable
* Reviewer J68b highlighted novelty and timeliness.
* Reviewer JvPM praised novelty, impact, and experimental thoroughness.
* Reviewer Ks1c initially had concerns about experimental details (e.g., optimizer, rank computation) but was satisfied with clarifications and commented that they will raise the score to positive.

During the rebuttal the following points came up where we added additional clarification.
1. Reviewer ks1c had concerns regarding the limited explanation of some experimental settings and on how the rank of the updates was computed
    * We offered a detailed explanation of how the rank was computed, expanding on the description in lines 174–179 of the main paper. In addition, we provided further information on the hyperparameters. While consolidating all hyperparameters into a single section is challenging given the nature of the paper, we will include section-specific details in the appendix of the revision.
2. How does the choice of optimizer impact this observed phenomenon of sparse updates? What is the effect of weight decay and learning rate ?
    * The choice of optimizer does not appear to eliminate the phenomenon of sparse updates. In particular, we trained a DPO model with RMSProp and observed that sparse updates still occurs to empirically support our argument. Similarly, experiments varying weight decay yielded consistent results, with us providing a detailed rationale for this behavior. Regarding learning rate, we noted that using a value higher than standard practice destabilized training and led to substantial performance degradation—an outcome consistent with both prior work and the authors’ own rebuttal experiments.
3. Reviewer J68b suggested rephrasing a few sentences in the main paper, and we outlined how these would be updated accordingly. J68b found the revisions thoughtful and retained their original score of 5.

We thank the reviewers again and hope that we were able to address their concerns.

---

### Decision · Program_Chairs · 2025-09-17

**Decision:**

Accept (poster)

**Comment:**

This work provides an empirical study revealing that RL training for LLMs updates only a sparse subset of the full model, leaving the rest unchanged. They conducted numerical verification to compare the difference between open-source models w/ and w/o RL training. Besides, they conducted experiments to compare full-parameter RL training and subnet RL training; their results show subnet training can achieve similar results on many benchmarks and even better performance than the full-parameter training. They conduct ablation studies to investigate the cause for this phenomenon and attribute that training on in-distribution data is a major reason of sparsity.

All reviewers appreciate the novel insights found in this paper. The AC agrees and thus recommends acceptance.